**Key words:**
Basic helix–loop–helix transcription factors; DNA deformation; DNA–protein interactions; torsionally-stressed DNA; molecular dynamics simulations

**Author for correspondence:**
*Anna Reymer,
E-mail: anna.reymer@gu.se

# Homologous basic helix–loop–helix transcription factors induce distinct deformations of torsionally-stressed DNA: a potential transcription regulation mechanism

Johanna Hörberg, Kevin Moreau and Anna Reymer*

Department of Chemistry and Molecular Biology, University of Gothenburg, 40530 Gothenburg, Sweden

## Abstract

Changing torsional restraints on DNA is essential for the regulation of transcription. Torsional stress, introduced by RNA polymerase, can propagate along chromatin facilitating topological transitions and modulating the specific binding of transcription factors (TFs) to DNA. Despite the importance, the mechanistic details on how torsional stress impacts the TFs-DNA complexation remain scarce. Herein, we address the impact of torsional stress on DNA complexation with homologous human basic helix–loop–helix (BHLH) hetero- and homodimers: MycMax, MadMax and MaxMax. The three TF dimers exhibit specificity towards the same DNA consensus sequence, the *E*-box response element, while regulating different transcriptional pathways. Using microseconds-long atomistic molecular dynamics simulations together with the torsional restraint that controls DNA total helical twist, we gradually over- and underwind naked and complexed DNA to a maximum of ± 5°/bp step. We observe that the binding of the BHLH dimers results in a similar increase in DNA torsional rigidity. However, under torsional stress the BHLH dimers induce distinct DNA deformations, characterised by changes in DNA grooves geometry and a significant asymmetric DNA bending. Supported by bioinformatics analyses, our data suggest that torsional stress may contribute to the execution of differential transcriptional programs of the homologous TFs by modulating their collaborative interactions.

## Introduction

Changing torsional restraints on DNA comprise one of the major regulatory forces of eukaryotic transcriptional control (Lavelle, 2008; Kouzine *et al.,* 2013; Ma *et al.,* 2013; Naughton *et al.,* 2013; Corless and Gilbert, 2016; Fogg *et al.,* 2021). In transcription, torsional strain is primarily introduced by RNA polymerase which forces DNA to rotate around its axis as the molecule threads through the transcription machinery (Osborne *et al.,* 2004; Boeger *et al.,* 2005). The imposed torsional strain underwinds and overwinds DNA upstream and downstream of a transcribed gene, respectively, and can propagate along the chromatin fibre. The speeds and ranges of torsional strain propagation depend on the underlying nucleotide sequence (Naughton *et al.,* 2013). When propagating along the chromatin fibre torsional stain brings changes to the genome organisation: locally introducing DNA bending and twisting, which affects the stability of nucleosome core particles (Teves and Henikoff, 2014), and higher-order chromatin structures (Corless and Gilbert, 2016). Through these changes, torsional strain can modulate transcription of near-located genes. Generally, genes that experience torsional strain are more efficiently transcribed (Weintraub *et al.,* 1986; Dunaway and Ostrander, 1993). Furthermore, the torsional strain might be necessary for the initiation of transcription (Tabuchi and Hirose, 1988; Mizutani *et al.,* 1991*a*, 1991*b*; Schultz *et al.,* 1992; Dunaway and Ostrander, 1993), as DNA underwinding appears essential for the melting of the TATA-box sequence near transcription start sites (Liebl and Zacharias, 2020). To complete the picture of how torsional strain contributes to eukaryotic transcriptional control, we must also understand how torsional strain impacts DNA specific binding by transcription factor proteins (Noy *et al.,* 2016; Hörberg and Reymer, 2020; Pyne *et al.,* 2021).

Locally DNA responds to moderate torsional strain, below any buckling transitions (i.e. plectonemes formation or melting) (Lankaš, 2020; Ott *et al.,* 2020; Dohnalová and Lankaš, 2022), in a heterogeneous and sequence-specific manner. Certain dinucleotide steps, mainly pyrimidine–purine (YpR) but also purine–purine (RpR), depending on their flanking environment exhibit twist bimodality, where two substates can be separated by as much as 20° in twist (Kannan *et al.,* 2006; Liebl and Zacharias, 2017, 2020; Hörberg and Reymer, 2018; Reymer *et al.,* 2018). These flexible dinucleotides, termed 'twist-capacitors', contribute to a multi-well free energy surface of DNA twisting, can effectively absorb torsional strain allowing the rest of DNA to preserve a B-like conformation. The twist-capacitor dinucleotides appear to regulate

protein-DNA complexation, as twisting transitions are coupled to changes in shift and slide (Dans *et al.,* 2012, 2019; Pasi *et al.,* 2014; Balaceanu *et al.,* 2019) – helical parameters important for the protein-DNA readout mechanism (Hörberg *et al.,* 2021). Previously, we addressed the impact of torsional strain on DNA complexation with a human basic-leucine-zipper (BZIP) transcription factor (TF) MafB (Hörberg and Reymer, 2020). When specifically bound to its DNA target, the protein locks the twist-capacitor dinucleotides in one conformational substate, favourable for the complexation. Consequently, the energy cost for DNA twisting almost doubles – suggesting that BZIP factors may hinder the propagation of torsional strain along DNA, potentially regulating the cooperative binding of collaborative TFs, or contributing to alterations in genome topology. These results provide first insights; a complete understanding of the mechanistic aspects of how torsional strain affects TF-DNA complexation remains limited. TF-DNA complexes involving other families of TFs may respond differently to torsional stress. Also, there might be differences among homologous TFs.

We herein address the impact of torsional strain on DNA complexation with three homologous basic helix–loop–helix (BHLH) TFs dimers, MycMax, MadMax and MaxMax. The BHLH factors is one of the most abundant families of eucaryotic TFs that modulate cell proliferation, differentiation and apoptosis (Atchley and Fitch, 1997; De Masi *et al.,* 2011; Dennis *et al.,* 2019; de Martin *et al.,* 2021). The MycMax, MadMax and MaxMax dimers exhibit specificity towards identical DNA consensus sequences – yet regulate different transcriptional pathways (Grandori *et al.,* 2000; Diolaiti *et al.,* 2015; Giardino Torchia and Ashwell, 2018). We study MaxMax, MycMax and MadMax dimers bound to the palindromic sequence '5′-GGCGAGTAG*CACGTG*CTACTCGC-3', containing the *E*-box response element (in italics) under relaxed and torsionally restrained conditions using microsecond-long umbrella sampling molecular dynamics (MD) simulations. With the torsional restraint that controls the total twist of DNA, that is, the end-to-end twist, without restricting other degrees of freedom, we gradually over- and underwind DNA to a maximum of $\pm$ 5°/base pair (bp) step. We observe that, relative to unbound DNA, the BHLH factors make DNA more torsionally rigid, resulting in similar torsional moduli for complexed DNA. However, under torsional stress, the BHLH factors induce distinct DNA deformations. We complement our mechanistic studies with bioinformatics analysis. Using CHIP-seq data on several cell lines, we explore what other TFs bind DNA in close proximity to Myc, Max and Mad proteins. Our results suggest: distinct responses to DNA twisting by homologous TFs may be a complementary mechanism to variations in transactivation domains (TAD), contributing to the recruitment of different collaborative TFs and, subsequently, differential regulatory responses.

## Methods

### Simulated systems

Four systems: MycMax- (PDB ID: 1NKP) (Nair and Burley, 2003), MadMax- (PDB ID: 1NLW) (Nair and Burley, 2003), MaxMax-bound DNA (PDB ID: 1AN2) (Ferré-D'Amaré *et al.,* 1993) and free DNA in B-form were studied. All systems contain a DNA 23-mer: GGCGAGTAG*CACGTG*CTACTCGC, with the *E*-box region in italics. USCF Chimera (Pettersen *et al.,* 2004) was used to modify the DNA sequence of the MaxMax-DNA complex to match that of the MycMax-DNA and MadMax-DNA complexes; and to

determine the protonation state of His residues. JUMNA (Lavery *et al.,* 1995) program was then used to extend the flanking sites and to relax bad protein-DNA contacts.

### Molecular dynamics simulation protocol

All molecular dynamics (MD) simulations are performed with the MD engine GROMACS v2019.4 (Abraham *et al.,* 2015) using same protocol as described previously (Hörberg and Reymer, 2020). In the restrained MD simulations of cascade umbrella sampling, we use the in-house developed torsional restraint (Reymer *et al.,* 2018) that controls the end-to-end twist (total twist) of a DNA fragment. The code of the DNA twisting restraint and a user guide is available at https://github.com/annareym/PLUMED_DNA-Twist. The restraint sets the desired value of *twist*$_{ref}$ using a simple quadratic function, $K_{tw}$ (twist–twist$_{ref}$)$^2$, implemented via PLUMED v2.5.3 (Bonomi *et al.,* 2009). We use the force constant ($K_{tw}$) of 0.06 kcal mol$^{-1}$ degree$^{-2}$, the smallest value that provides the desired twist without inducing any structural artefacts. In all simulations we use AMBER 14SB (Maier *et al.,* 2015) and Parmbsc1 (Ivani *et al.,* 2016) force fields to treat the proteins and DNA, respectively. The protein-DNA complexes and free DNA oligomer are separately solvated in cubic periodic boxes by SPC/E (Mark and Nilsson, 2001) water molecules with a buffer distance of 12 Å to the walls. The systems are first neutralised by K+ counterions, then additional K+ and Cl− ions are added to reach a physiological salt-concentration of 150 mM. Applying periodic boundary conditions, each system is subjected to energy minimization with 5,000 steps of steepest descent, followed by 500 ps equilibration-runs with week position restraints on heavy solute atoms (1,000 kJ mol$^{-1}$) in the NVT and NPT ensembles to adjust temperature (Berendsen *et al.,* 1984) and pressure (Parrinello and Rahman, 1981) to 300 K and 1 atm. Releasing the restraints, 0.6 μs simulations are then carried out at constant pressure and temperature (1 atm and 300 K).

Following the unrestrained MD simulations, the cascade umbrella sampling (Torrie and Valleau, 1977) is performed with 0.5 μs sampling time per umbrella window to allow sufficient convergence of DNA conformational substates and ion populations. We apply the twist restraint to the central *E*-box region and the four adjacent 5′- and 3′-flanking nucleotides; 13 bp steps in total (GGCGA*GTAGCACGTGCTAC*TCGC). The initial value of twist$_{ref}$ is the averaged end-to-end twist of the restrained fragment for each system, obtained through postprocessing of the unrestrained MD simulations with the twist restraint. For unbound-DNA, MycMax- and MaxMax-bound DNA the value of twist$_{ref}$ is initially set to 450°, and for MadMax-bound DNA – to 445°. Starting from a relaxed state, the end-to-end twist of the restrained fragment is gradually changed by $\pm$ 0.5°/bp step ($\pm$6.5° in total per umbrella window), until a maximum overwound and underwound state of 5°/bp step is reached. The final structure from every window is used as the starting point for the following umbrella window. The weighted histogram analysis method (WHAM) (Kumar *et al.,* 1992), implemented in PLUMED is used to derive the potential of mean force (PMF) with respect to DNA twisting. The total umbrella sampling MD simulation time for each system is 10.5 μs.

### Elastic force constant analysis

Quadratic regression analysis in MatLab is used to obtain the twisting force constants, K (kcal mol$^{-1}$ deg$^{-2}$) from the PMF profiles. The analysis is performed for the regions corresponding to a Δtw of $\pm$ 2°. The derived force constants are used to calculate

DNA torsional modulus at room temperature in nm according to the homogeneous rod model, given by Eq. (1); where $T$ is the torque that results from a change in twist $\Delta\theta$ over a standard bp length $L$ (0.34 nm):

$$T = K\Delta\theta = \frac{C\Delta\theta}{k_B\,TL}. \tag{1}$$

### Trajectory analysis

Curves+ and Canal (Lavery *et al.,* 2009) programs are used to derive DNA helical parameters, backbone torsional angles, and groove geometry parameters for each trajectory snapshot extracted at 1 ps intervals. DNA deformation energies for the restrained region (GGCGA*GTAGCACGTGCTAC*TCGC) are calculated using a model by Liebl and Zacharias (2021). The model combines a quadratic harmonic deformation approximation model with an Ising model to allow for inclusion of coupling between all possible conformational substates of the DNA duplex. The model has been parameterized for all 136 tetranucleotides. The model utilises the six inter-base pair (shift, slide, rise, twist, tilt and roll) and the six intra-base pair (shear, stagger, stretch, buckle, propeller twist and opening) parameters to calculate the deformation energy for DNA. The deformation energies have been calculated for every snapshot for the relaxed, $\pm 4.5°$/bp, and $\pm 2.5°$/bp twisting trajectories, which provides the trajectory average DNA deformation energy and the standard deviations. The CPPTRAJ (Roe and Cheatham, 2013) tool from AMBERTOOLS16 software package is used to derive specific and nonspecific protein-DNA contacts along the different torsional regimes as described previously (Hörberg and Reymer, 2020).

### Bioinformatic analysis

Analysis of the nearest binding partners of Myc and Max was performed using TFregulomeR (Lin *et al.,* 2020) package on ChIP-seq datasets from multiple cell lines (A549, NB4 and K562) downloaded from TFregulomeR data compendium. The output gives, among other information, the top-10 co-binders (ChIP-seq signal around $\pm 100$ bp of the peaks summits, corresponding to the binding of the studied TFs). The TFregulomeR package allows to differentiate the Myc and Max binding distribution corresponding district genomic regions such as promoter, introns, exons, intergenic regions, and so forth.

### Additional information

MatLab and R software (R Core Team, 2013) were used for postprocessing and plotting of all data. USCF Chimera (Pettersen *et al.,* 2004) was used for creating molecular graphics.

## Results

### Torsional moduli of free and complexed DNA

To address how changing torsional restraints on DNA impact the molecular complexation with homologous transcription factors and consequently their transcription regulatory programs, we select homo- and heterodimers of the Myc/Max/Mad network. The Myc/Max/Mad proteins belong to the BHLH family of eukaryotic transcription factors (TFs) that exhibit specificity towards the same DNA response elements but play distinct roles in transcriptional control. Upon association with its DNA target sites, the MycMax dimer acts as a transcriptional activator, which induces histone acetylation. The MadMax dimer antagonises MycMax and acts as a transcriptional repressor, which recruits histone deacetylases. The MaxMax dimer also antagonises MycMax, however, since Max lacks a transactivation/transrepression domain, it is considered transcriptionally inert (Grandori *et al.,* 2000).

We first subject the MaxMax-, MycMax- and MadMax-DNA complexes and free DNA, containing the E-box response element sequence ('GGCGAGTAG*CACGTG*CTACTCGC') (Fig. 1) to unrestrained MD simulations. We continue with cascade umbrella sampling using the torsional restraint that controls end-to-end twist of a restrained DNA region. The restraint is applied to the E-Box response element and the four adjacent 5′- and 3′ flanking nucleotides, 13 bp steps in total. Starting from the relaxed duplex, we gradually over- and underwind free and protein-bound DNA until reaching a maximum of $\pm 5°$/bp step (corresponding to $\pm 0.15$ in supercoiling density, $\sigma$). In accordance with previous studies, at all simulated torsional regimes the restrained DNA region in the four systems preserves a *B*-like conformation. We observe no significant DNA bp flipping or melting, as supported by DNA stretch and opening distributions (Supplementary Figs 3A–D and 4A–D) as well as by hydrogen bond distances between the heavy atoms of Watson–Crick bps, which remain below 4 Å (Supplementary Fig. 5). In addition, we observe no significant bending for free DNA (Supplementary Fig. 2) with a smooth decrease in the bending angle when going from underwound to overwound state. For protein-bound DNA, bending becomes more significant at higher degrees of underwinding and overwinding (Supplementary Fig. 2), reflecting changes in DNA groove geometry and roll angles due to the protein presence.

The selected torsional restraints range should be seen as an approximation of extreme local changes that may arise, for example, near transcription starting sites upon transcription initiation (Naughton *et al.,* 2013; Irobalieva *et al.,* 2015; Muskhelishvili and Travers, 2016), which allows us to gain mechanistic insights into these highly dynamic aspects of eukaryotic transcriptional control. The selected range may be an exaggeration, as the value of $\sigma = \pm 0.15$, is much higher than the upper limit of negative supercoiling density that has been measured ($\sim -0.07$) in the nucleus. However, the value of $\sigma = -0.07$ constitutes an average for bulk DNA, while DNA supercoiling density is not uniformly distributed in the genome, it can vary significantly along genomic DNA and also vary rapidly with time (Naughton *et al.,* 2013). Particularly, for transcription initiation, local negative DNA supercoiling density could in principle reach a value of $-1$ (Muskhelishvili and Travers, 2016), as DNA undertwisting facilitates the formation of the transcription bubble. The bubble is concentrated in short sequence regions ~10–20 bp, where no writhing is expected to occur and changes in DNA supercoiling density will be accommodated through changes in bp twist. Furthermore, the net supercoiling density generated by RNA polymerase is zero, thus the local change in positive supercoiling density should be equal to that of negative supercoiling density. As DNA complexes with TF proteins are formed in the vicinity of transcription starting sites, they will therefore experience the full range of extreme changes in DNA supercoiling density, which justifies the selected range of the torsional restraints. However, the ultimate range of *in vivo* changes in torsional restraints on DNA remains to be determined.

From the torsionally restrained MD simulations we derive the potential of mean force (PMF) profiles showing the free energy cost for DNA twisting transitions (Fig. 2*a* and Supplementary Fig. 1). To compare the PMF profiles, we plot the changes in the end-to-end

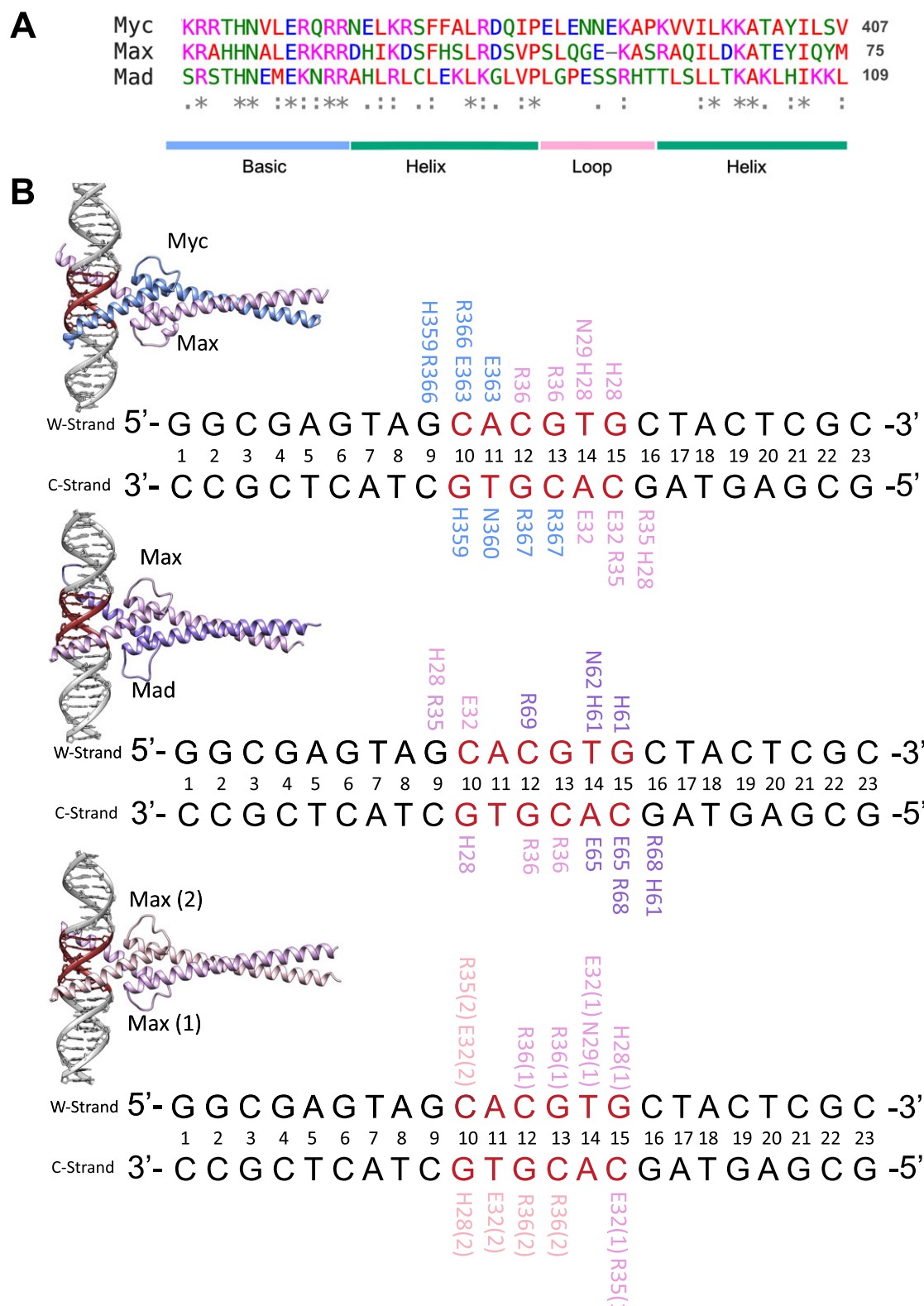

**Figure 1.** (*a*) Sequence alignment of the basic helix–loop–helix domain of Myc, Max and Mad. (*b*) Crystal structure of MycMax-DNA (PDB ID: 1NKP), MadMax-DNA (PDB ID: 1NLW) and MaxMax-DNA (PDB ID: 1AN2) complexes bound to the E-Box element in red. For each complex, specific protein-DNA contacts seen in the crystal structures are highlighted. The Watson strand (5′– > 3′) is denoted with '*w*' and the Crick (3′– > 5′) strand is denoted with '*c*'.

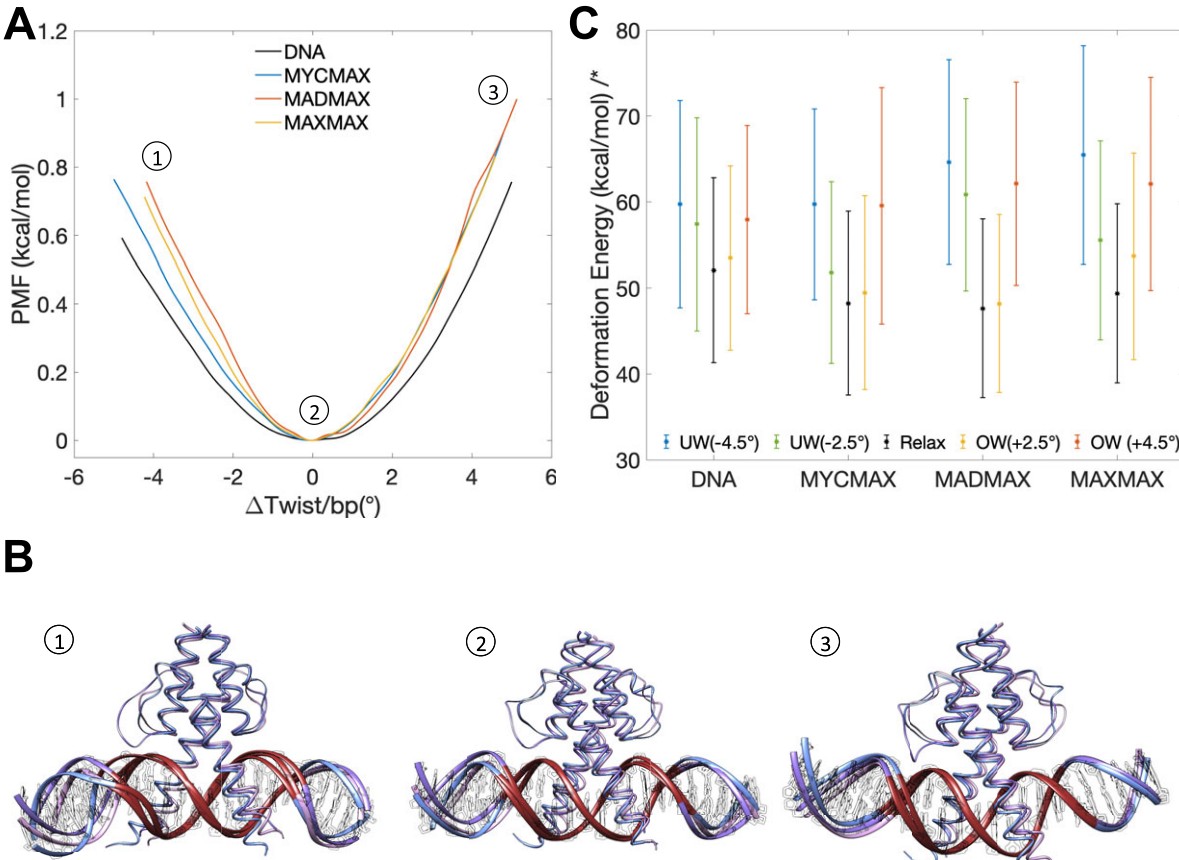

**Figure 2.** (*a*) PMF profiles showing the energy cost for twisting transitions for free and BHLH-bound DNA. (*b*) MycMax-, MadMax- and MaxMax induced DNA deformations are shown for (1) underwound regime, (2) torsionally relaxed regime and (3) overwound regime; with the restrained DNA region in red, MycMax in blue, MadMax in magenta, and MaxMax in pink. (*c*) DNA deformation energy of the restrained region (GTAG*CACGTG*CTAC, E-box response element in italics) calculated with a multivariate Ising model (Liebl and Zacharias, 2021).

twist as the average twist per bp step with respect to the relaxed average twist that varies from 34.0° for MadMax-bound DNA to 34.7° for MycMax-bound DNA (Supplementary Table 1). To compare the torsional rigidity of DNA in the four systems, we derive the torsional force constants and torsional moduli, using quadratic regression (Supplementary Fig. 6 and Supplementary Table 1). We observe that the binding of the BHLH dimers make DNA more torsionally rigid, resulting in the torsional moduli of 153, 163 and 168 nm for MycMax-, MadMax- and MaxMax-bound DNA, respectively, *versus* 111 nm for free DNA. The observation is consistent with our previous study of a BZIP factor MafB (Hörberg and Reymer, 2020), where we showed that the TF binding almost doubles the torsional rigidity of DNA (207 nm for complexed *versus* 107 nm for free DNA). It should be noted that MafB recognises longer response elements (13–14 bps) compared to the BHLH factors (6 bps), which may explain the differences in the induced torsional rigidity of TF-bound DNA. Though, other aspects of TF-DNA complexation, for example, whether proteins form specific contacts with twist-capacitor dinucleotides. Nevertheless, the increased torsional rigidity of TF-bound DNA, suggests that TFs may hinder the propagation of torsional stress. However, we believe, the effect is local and by increasing the length of the restrained region the torsional modulus for TF-bound DNA will eventually converge towards that of free DNA. Yet in the topological conditions of real genomes, the local increase in DNA torsional rigidity may provide a substantial regulatory force.

Despite the similar increase in DNA torsional rigidity, induced by the protein binding, we observe that the three BHLH dimers deform DNA in a different fashion. Starting from − 1.0°/bp during the underwinding regime, we observe that DNA molecules experience a local bending deformation induced by an increased roll angle at several bp steps and changed geometry of DNA grooves (Fig. 2*b*, Supplementary Fig. 1, and Supplementary Videos 1–4). The torsionally induced deformations differ also between free and protein-bound DNA (Supplementary Fig. 1). For free DNA the imposed torsional stress is evenly distributed over the entire restrained region (Fig. 3), with no significant bending even at higher torque regimes (Supplementary Fig. 2). While for protein-bound DNA (see details below), the imposed torsional stress is mostly accumulated in the flanking regions outside the *E*-box response element (Fig. 3), where the observed deformations are predominantly localised.

For a subset of twisting trajectories, we also calculate the deformation energies (Fig. 2*c*) for the restrained DNA regions (GGCGA*GTAGCACGTGCTAC*TCGC) using a model by Liebl and Zacharias (2021), which allows to estimate the change of free energy coupled to changes in DNA conformational flexibility due to the imposed torsional stress. The deformation energies have been averaged over the entire umbrella window (500 ns) for the corresponding degrees of under- and overwinding. Although the standard deviations are rather high (~ 10 kcal mol$^{-1}$), illustrating the large conformational landscape of DNA, the trend shows that in all

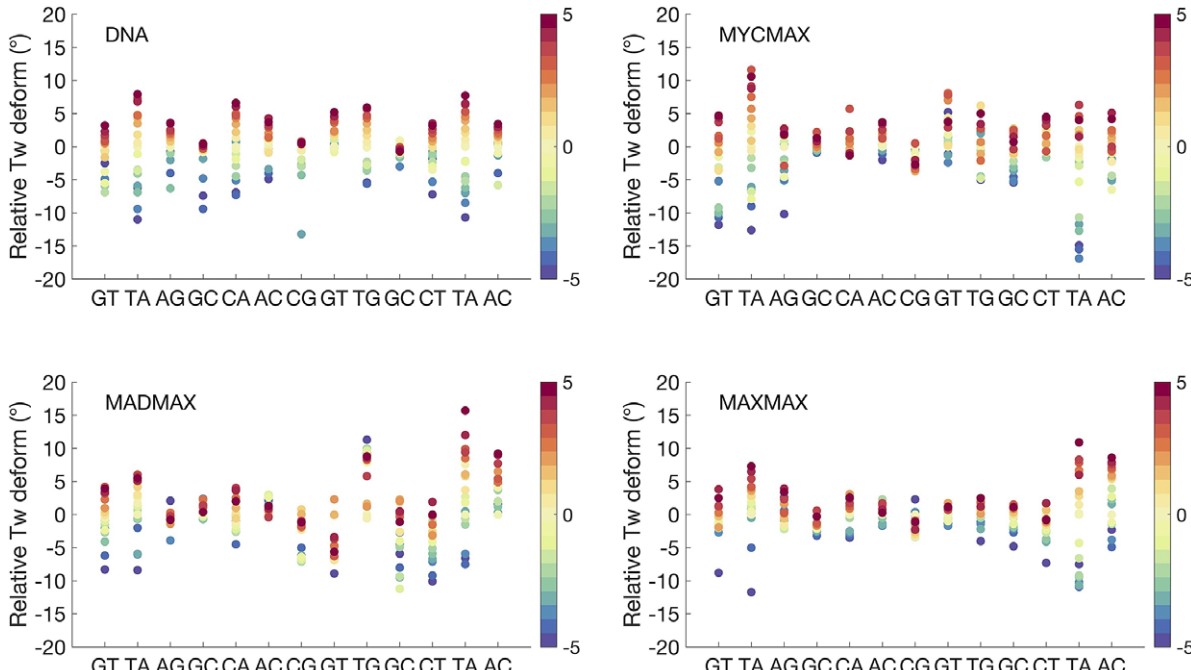

**Figure 3.** Changes of bp twist angles for the restrained DNA-region in free and MycMax-, MadMax- and MaxMax-bound DNA as a function of the requested average change of twist per base pair step, indicated by a colourbar to the right.

four systems the deformation energies increase as the torsional restraints are applied, both for under- and overtwisting. The increase in energy is explained by the fact that the torsional stress shifts the distribution of DNA conformational substates, allowing the less populated substates at relaxed conditions to become more populated; these transitions cost more energy. The trends in Fig. 2c also show that the increase in deformation energies is system-specific, suggesting that torsional stress may modulate the differential transcriptional response of homologous BHLH TFs by differently impacting the local conformational flexibility of DNA.

### Torsional stress-induced changes in DNA structure

To understand the BHLH dimer-specific torque-induced DNA deformations, we first analyse the contributions of individual bp steps to the absorption of the imposed torsional stress (Fig. 3 and Supplementary Fig. 7). We exclude from the discussion the outer bp steps (GpT and ApC) of the restrained region (GTAG*CACGTG*CTAC, the *E*-box response element in italics), as these steps absorb significant amount of torsional stress at the extreme torque regimes, which may be attributed to their location (on the edge of the restrained region) rather than any other effects. Contrary to free DNA, where the imposed torsional stress is absorbed by the flexible pyrimidine–purine steps, TpA and CpA, of the palindrome (*GTAGCACGTGCTAC*), for protein-bound DNA the CpA and TpG steps within the *E*-box sequence become torsionally-rigid. As a result, the torque accumulates mostly in the flanking regions. For MycMax- and MaxMax-bound DNA, the flanking TpA steps dominate the absorption of both negative and positive torque (Fig. 3). However, as the TpA steps exhibit a preference for a high twist state under the relaxed conditions (Supplementary Fig. 7), they are less efficient in absorbing positive torsional stress, contributing to an increased energy cost for DNA overtwisting. For MadMax-bound DNA the first flanking TpA step remains rigid, while the second flanking TpA step efficiently

absorbs only positive torsional stress as it prefers a low twist state under the relaxed conditions (Fig. 3 and Supplementary Fig. 7). Thus, the imposed negative torsional stress is distributed among the less torsionally flexible bp steps in the second half-site of the restrained DNA region, which increases the energy cost for DNA undertwisting. Interestingly, for MadMax-bound DNA the ApG step (GT*AG*CACGTGCTAC) appears undergoing non-monotonic changes upon high level of undertwisting (−4.0°/bp step), that is, its twist increases. This will be further discussed in the next section, as this behaviour is due to the protein-DNA contacts formed by the loop of the Mad factor, which makes the TpA step torsionally rigid. However, upon high undertwisting, the increase in twist of the ApG step allows the TpA step to absorb some negative torsional stress, while still maintaining the interactions with the Mad-loop.

For the torsionally active bp steps, behaviour of other translational and rotational bp steps parameters (Supplementary Figs 8–14) under torsional stress follow the trends reported in our previous studies (Hörberg and Reymer, 2018; Reymer *et al.,* 2018): torsional stress brings changes in roll, shift, and slide – which are coupled to twist via BI/BII backbone conformational transitions. Roll shows negative correlation to twist, and slide – positive. Behaviour of the shift parameter appears more complexed. In the relaxed state, we observe shift bimodality/multimodality for the torsionally flexible bp steps, which changes as DNA undergoes under- and overtwisting.

We further characterise the differences in the BHLH dimer-specific torque-induced DNA deformations, by analysing DNA axis bending per bp step, characterised as the angle between the local axes of two adjacent bps (Fig. 4) and groove parameters (Fig. 5). For MycMax- and MaxMax-bound DNA we observe an asymmetric bending towards the major groove, that is, towards the basic region of the BHLH factors, during the underwinding regime. The axis bending for MycMax- and MaxMax-bound DNA, from the Myc- and Max_1-side, respectively, gradually increases with undertwisting up to 5° per bp step (Fig. 4 and Supplementary

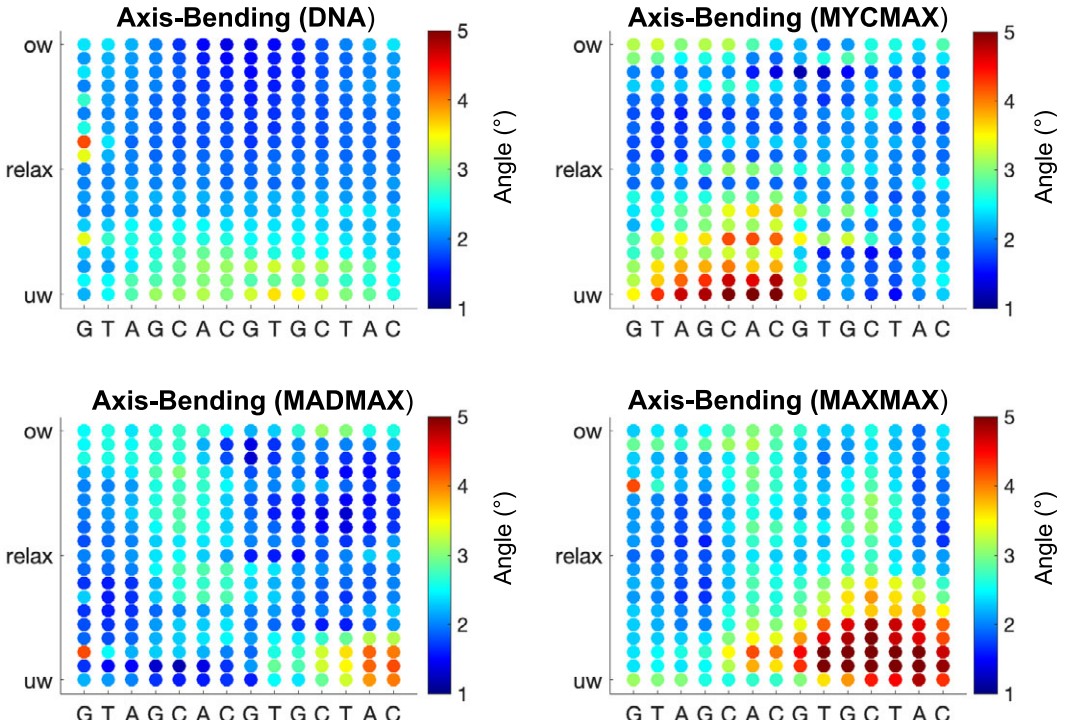

**Figure 4.** Change in average axis bending of bp of the restrained DNA region (GTAGCACGTGCTAC) along the torsional regimes denoted with a colourbar.

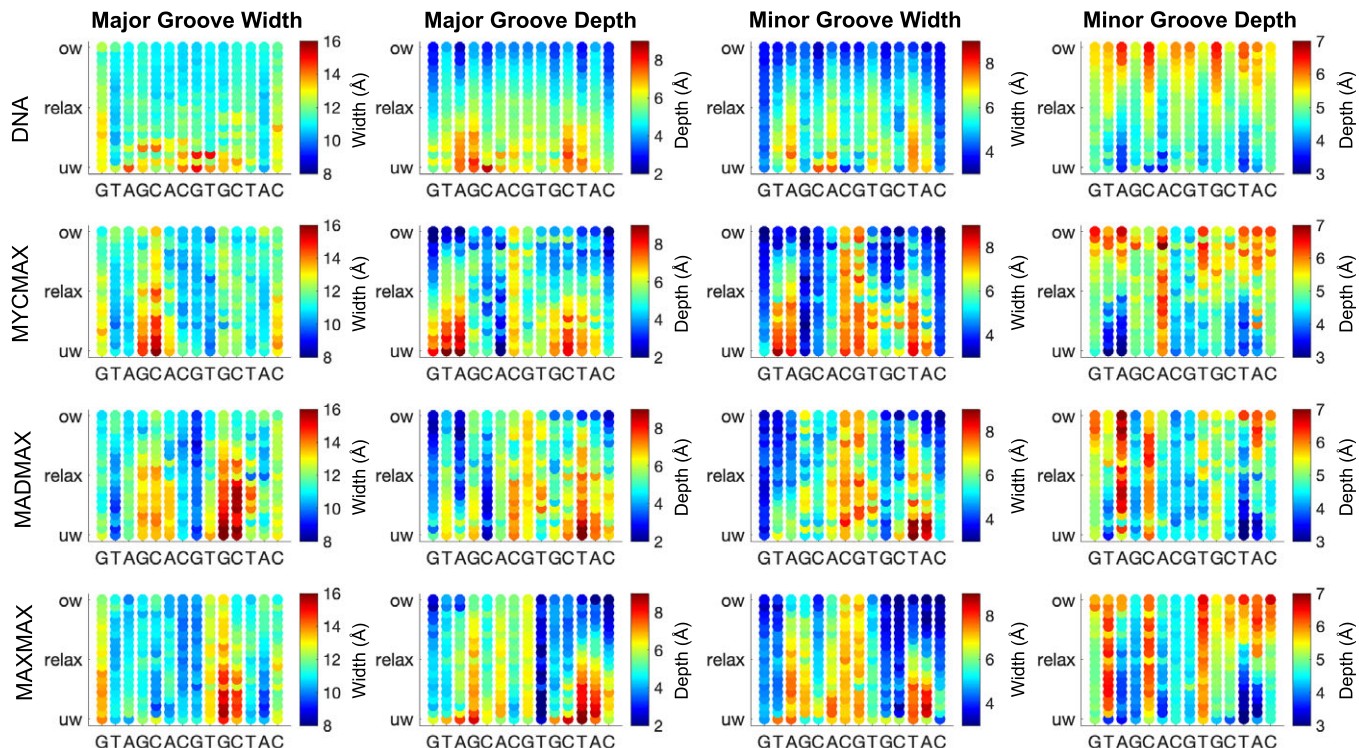

**Figure 5.** Change in average depth and width of major and minor grooves of the restrained DNA region (GTAGCACGTGCTAC) along the torsional regimes denoted with a colourbar.

Videos 2 and 3). Max_1 of the MaxMax homodimer refers to the monomer, which forms a greater number of specific contacts with DNA (see Protein-DNA contacts for further details). Contrary, for MadMax-bound DNA, similarly to free DNA, we observe no significant DNA bending during underwinding. Overwinding also

contributes to an increase in axis bending for all BHLH-bound DNA. However, the changes are smaller, about 2° per bp step, and more uniformly distributed over the restrained region, resulting in a symmetric smooth bending towards the minor groove, that is, towards the BHLH-loops (Supplementary Fig. 1 and Supplementary

Videos 2 and 3). For DNA groove parameters, underwinding results in an increase in the major groove width and depth, and a decrease in the minor groove depth. The reverse trend is observed during overwinding. For BHLH-bound DNA, changes in both grooves are more noticeable for the flanking regions that accumulate most of the imposed torsional stress (Fig. 3), that is, the flanking sequences of the Myc-, Max_1-, and Mad sides, respectively.

### Protein-DNA contacts networks at different torsional stress regimes

We continue with the analysis of differences in the intermolecular contacts networks exploited by the three BHLH dimers. To recognise their DNA targets, the BHLH family utilises a five-residues motif (**xx*E*xx*R*) (De Masi *et al.,* 2011). For Myc, Mad, and Max

the five-residues motif corresponds to *HNxxExxRR* (Fig. 1). Upon DNA binding, one of the monomers of the BHLH dimers forms further specific contacts with the *E*-box sequence, this includes Myc of MycMax, Mad of MadMax, and Max_1 monomer of MaxMax (Figs 1 and 6 and Supplementary Figs 16A–C and 17A–C). The other monomer interacts with DNA more nonspecifically. The Myc/Mad/Max_1 monomer shows nearly identical networks of specific intermolecular contacts (Fig. 6 and Supplementary Fig. 16A–C). In the torsionally relaxed state, from the five-residues motif (HNxxExxRR), histidine interacts specifically with the TG bp step on the opposite strand of the *E*-box half-site (CACG/CG*TG*); asparagine with the T base on the opposite strand of the *E*-box half-site (CACG/CG*T*G), glutamate with the CA bp step (*CA*CG/CGTG) and the T base on the opposite strand of the *E*-box half-site (CACG/CG*T*G); first arginine with

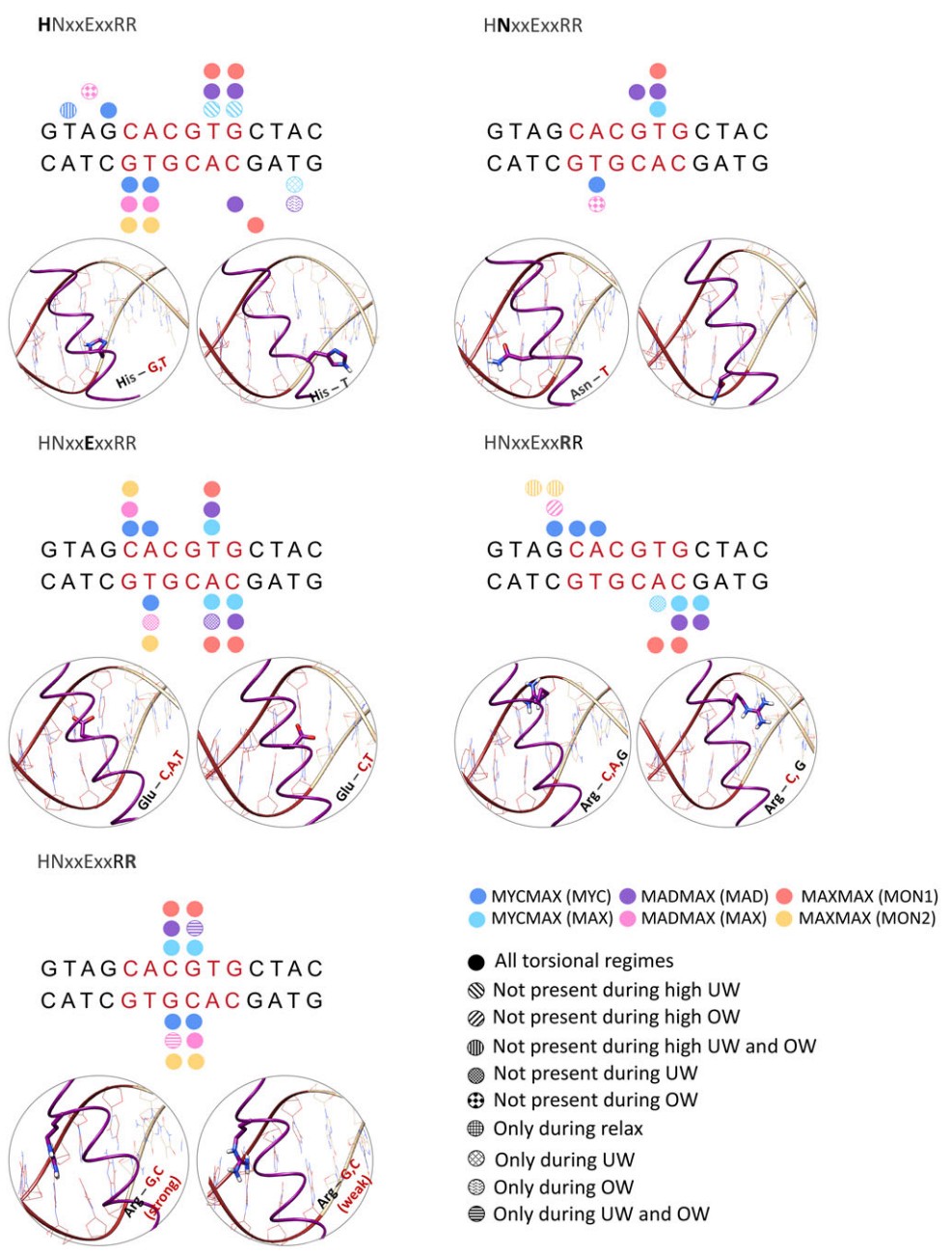

**Figure 6.** Specific contacts exploited by the five residues motif HNxxExxRR during the different torsional regimes.

the CA bp step (*CA*CG/CGTG) and the flanking sites; and second arginine with the central CG bp step on the opposite strand of the *E*-box half-site (GCACG/*CG*TG). Furthermore, the MadMax dimer has an additional specific contact, the Arg91 residue of the Mad loop interacts with the flanking TA step (G*TA*GCACG) from the minor groove (Supplementary Figs 16 and 17B), which explains the torsional rigidity of the step.

The intermolecular specific interactions by the Max monomer, including Max of the MycMax and MadMax heterodimers, and Max_2 monomer of the Max-homodimer, follow to some extend the above-described specific contacts (Fig. 6 and Supplementary Figs 16 and 17). The differences include, for Max of MadMax and Max_2 of MaxMax, the Glu residue shows no interactions with the CA bp step; the first Arg residue interacts only with the flanking sites; for the MadMax dimer the second Arg residue oscillates between specific and nonspecific interactions with the CG bp step (Supplementary Fig. 16).

We further analyse the evolution of the strength of specific and nonspecific contacts, and the total number of protein-DNA contacts along the different torque regimes (Supplementary Figs 15 and 16). The analyses show that most of the protein-DNA contacts remain stable over the different degrees of torsional strain. There are, however, BHLH dimer- and torque-specific differences in the protein-DNA contacts that can explain the observed differences in DNA torque-induced deformations during the underwinding regime. Upon high level of underwinding (>−3°/bp) the His residue (*H*NxxExxRR) of the Max monomer of the MycMax heterodimer rearranges its orientation to hydrophobically interact with the *T* base (step (GTAGCACGTGC*T*AC) Supplementary Fig. 17A), which limits the torsional activity of the corresponding TA step, leading to a smaller change in the roll angle and the major groove width. This in addition to a greater number of nonspecific interactions from the Max side creates potential steric clashes that hinders DNA bending. Similarly, during the underwinding regime the nonspecific interactions that involve the His27 and Leu31 residues of Max_2 in the MaxMax homodimer, and the Ser 55–57 residues of Mad in the MadMax heterodimer prevent DNA bending from the corresponding sites.

There are also torque-specific changes in the protein-DNA specific interactions that have no impact on the observed torque-induced BHLH dimer-specific DNA deformations. Those include the Glu-A specific contacts by Max of MycMax and Mad of Mad-Max that become weaker upon underwinding (Supplementary Fig. 16), the second Arg-G specific contact for Mad and Max in MadMax that becomes stronger upon underwinding.

We also observe fluctuations in the ratio of the specific to nonspecific contacts and their strengths, which are linked to the flickering power of the long side chain residues, oscillating between interactions with DNA bases and backbone. The overall stability of the intermolecular contact networks reflects that the BHLH factors form mainly contacts with the *E*-box response element, while the imposed torque is predominantly absorbed by the flanking regions. Nevertheless, there are also, as discussed, some differences in the contacts between DNA flanking sites and the different proteins, which impact the distribution of the torsional stress and the observed DNA deformations.

## Discussion

Our results show that homologous BHLH dimers, MycMax, Mad-Max and MaxMax can hinder the propagation of torsional stress

along the genome by making the bound *E*-box sequence more torsionally rigid. Consequently, the imposed torque is accumulated at the flanking sites, resulting in the distinct BHLH dimer-specific DNA deformations during the underwinding regime. The negative torque-induced deformations, characterised by changes in DNA grooves geometry and an asymmetric bending of the *E*-box half side flanks, are more significant for MaxMax and MycMax DNA. The deformations occur at the Myc and Max_1 side, as these monomers form more base-specific contacts than their dimer partners, Max and Max_2, respectively. Despite experiencing the distinct deformations, the increase in DNA rigidity is relatively similar (see Supplementary Table 1). Here, we want to point out that we study only one DNA sequence; it is likely that the observed DNA deformations are sequence specific. In addition, the studied DNA sequence is relatively short, for longer DNA the deformations may show a different amplitude as well as localisation.

Our results allow us to hypothesise that the torque-induced BHLH dimer-specific DNA deformations can contribute to the TFs differential transcriptional responses by producing binding sites for distinct collaborative proteins. To validate our hypothesis, we explore potential binding partners of the BHLH dimers, using the ChIP-seq data available for Myc and Max TFs, using a window of ±100 bp from the DNA binding sites of the proteins of interest. It should be noted that ChIP-seq data is not single-base resolved and the protein co-binding may occur anywhere from 1 to 100 bp. Thus, the identified binding partners may be affected if they bind in relative proximity to the analysed TFs. Alternatively, the torque-induced deformation may propagate along DNA in a domino-like fashion, where it can facilitate (or inhibit) binding of a neighbour protein that in turn will continue to deform DNA and affect protein binding of the further away sites. The analysis reveals that among ten most frequent binding partners there are members of E2F, BZIP, and Zinc finger TFs families (Supplementary Fig. 18A,B). Additionally, previous studies also listed the TATA-box binding protein (Wei *et al.,* 2019; Lourenco *et al.,* 2021), TBP, as a frequent co-binder of the MycMax dimer. Interestingly, the TBP and the E2F-factors deform DNA in a similar fashion as the one we observe for the restrained Max_1 and Myc-flanks of MaxMax- and Myc-Max-DNA, respectively, during the underwinding regime. Thus, potentially the binding of the MycMax/MaxMax dimers could facilitate the binding of the TBP and E2F factors. For BZIPs and Zinc fingers, which induce no major conformational change of DNA but due to the flexibility of their DNA binding domains may associate with significantly deformed DNA (Patel *et al.,* 2018; Hörberg and Reymer, 2020), the co-binding with MycMax/ MaxMax may bring other mechanistic advantages. Based on our study of MafB, we know that BZIP factors can also hinder the propagation of torsional stress along DNA. Thus, the tandem binding of BHLH and BZIP factors could further enhance the transient accumulation of torsional stress, which could be necessary for the destabilisation of nucleosome core particles, the pre-initiation complex formation (Corless and Gilbert, 2016), or DNA looping (Yan *et al.,* 2018). The analyses of Myc (Supplementary Fig. 18A) and Max (Supplementary Fig. 18B) show both similar and different co-binding partners, which relates to the fact that Max forms both homodimers and heterodimers.

In summary, using atomistic microsecond range umbrella sampling simulations with the torsional restraint that controls DNA total twist, we have shown that BHLH TFs may hinder the propagation of torsional stress along DNA. When complexed with the homologous MycMax, MadMax and MaxMax dimers, DNA show a similar increase in the torsional rigidity but experience distinct

torque-induced deformations, which may modulate the binding of collaborative TFs. We thus propose that changing torsional restraints on DNA may contribute to the differential transcriptional programs of homologous TFs.

**Acknowledgements.** The authors thank Swedish National Infrastructure for Computing (SNIC) for the generous provision of computing resources.

**Supplementary Materials.** To view supplementary material for this article, please visit http://doi.org/10.1017/qrd.2022.5.

**Data availability statement.** All data generated and analysed in this study are available from the corresponding author upon request.

**Author contributions.** J.H. and A.R. conceived and designed the study. J.H. performed the MD simulations. K.M. performed the bioinformatics analyses. J.H., K.M., and A.R. analysed the data. J.H. and A.R. wrote the article.

**Financial support.** This work was supported by the Swedish Foundation for Strategic Research SSF (grant number ITM170431).

**Conflict of interest.** The authors declare no conflicts of interest.

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
