## [Reviewer Report]

*Comments to Author*: This paper describes a fascinating new mechanism for supercoiled-controlled DNA recognition, whereby bound proteins trap supercoils, thereby changing the twist of the DNA upstream, and changing the affinity of this site for a second DNA binding protein. Simulations have been performed at various levels of DNA twist using restrained MD in the presence of three related DNA binding proteins, and the sequence dependent absorption of the twist imposed carefully assessed, thereby demonstrating how these different proteins impede supercoiling. The simulations performed have been carefully described and the analysis thoughtfully executed, with detailed information including error bars presented as Supplementary Information. The addition of Chip-Seq analysis suggests the potential for binding partners of the proteins investigated to co-operatively associate in a manner that can be controlled by supercoiling. I find this proposed mechanism very compelling, and therefore recommend publication of this interesting paper.

I have one recommendation to improve the readability of the paper. The authors may consider including the full proteins investigated in Figure 1, ideally highlighting differences in sequence, as currently the reader only gets to see this information when they reach the section on DNA protein contacts, or if they look at Supplementary Figure 2 (which explains this very clearly). They may also wish to carefully check the legend to Figure 1, as I found the numbering and lettering somewhat confusing.

---

## [Reviewer Report]

*Comments to Author*: Hörberg et al. study structural effects of torsionally deforming DNA double helix with different transcription factors (TFs) bound to it by means of extensive umbrella sampling molecular dynamics (MD) simulations. In the cell, DNA is torsionally stressed during transcription, which comprises a major regulatory force in transcription regulation, operating via different mechanisms. In this work the authors demonstrate that in the presence of different TFs, the torsional stress induces different local structural changes to the DNA. The authors then hypothesize that these differences might have functional implications for transcription control, thus representing yet another regulatory mechanism, and give hints to support this view.

The work proceeds much along the lines of the authors’ prior publication (Hörberg and Reymer, Sci. Rep. 2020), where the hypothesis has already been formulated. In the present manuscript, the authors apply their approach to three more protein-DNA complexes with important TFs. Another innovation is that now the torsionally restrained region is wider than the protein binding site (4 flanking base pairs at each side), so that the effect on the flanking DNA can also be monitored.

My main concern is the following. The authors apply their torsional restraint and see TF-dependent structural changes within the protein binding site and the +/- 4 flanking pairs. Then they perform bioinformatics analysis using ChIP-seq data and find that each TF has different co-binders within +/- 100 bp around its binding site. The authors then hypothesize that the two phenomena are related, in particular they propose that torque-induced, protein-specific DNA deformations produce binding sites for the distinct co-binders (p. 11). However, the authors do not propose any plausible physical mechanism whereby the localized structural changes propagate as far as 100 bp away. There seems to be nothing in their data pointing to this possibility. The authors then focus on a specific example, the TBP and E2F factors. They mention (p. 11) that TBP and E2F deform DNA in a similar fashion as the one they observe, but no details are given, so it is hard to understand what they mean. Notice that e.g. TBP binds to the TATA-box (TATAAAA and its variants) which is 7 bp long, longer than the 4 flanking pairs modelled here. The authors should clarify these points.

Other issues:

Multiple times throughout the manuscript: use "respectively" instead of "correspondingly". Also, pay attention to the proper form of citations in the text ("Hörberg and Reymer" instead of "Johanna Hörberg and Reymer", etc.)

P. 2: The discussion about twist-absorbing dinucleotides is incomplete. Notice that if an inhomogeneous rod is torsionally stressed (below its buckling point), then its more flexible parts absorb more twist. This can be understood already in the framework of linear elasticity, by computing minimum energy twist distribution under the constraint of fixed total twist. The possible multiwell free energy landscape may (or may not) contribute to this, but multimodality is certainly not a prerequisite for the phenomenon. Please refer e.g. to a recent review on DNA elasticity (Dohnalova and Lankas, Wiley Interdiscip. Rev. Comput. Mol. Sci. 2021).

P. 2, bottom: The authors promise to investigate the relative positions of the TFs with respect to the nearest nucleosomes and transcription start sites, but nothing of this kind seems to be present in the manuscript.

P. 2, 5th line from below: as well ◊ as well as

P. 3, 2nd para: The restrained sets … ◊ The restraint sets …

P. 3, 2nd para: degrees^{-2} ◊ degree^{-2}

P. 3, MD simulation protocol: My understanding of the authors’ method to impose twist is that three base pairs at each side of the examined region are chosen to define the restraint. Please say which base pairs were chosen. Also say from which to which pair you define the twist between the handles (end-to-end twist). Perhaps it is defined between the middle pairs of each handle?

P. 3: It is very important to indicate the value of twist_ref and how it was obtained. Was twist_ref the same for all systems?

P. 3: There seem to be 10 windows for undertwisting and 10 for overtwisting, plus one for twist = twist_ref. That would make 10.5 microseconds, but you talk about 11.1 microseconds, please explain. Use the proper symbol for a microsecond.

P. 3-4, Elastic force constant analysis: The authors use a quadratic fit in the vicinity of the minimum for undertwisting and overtwisting individually. This does not seem to be correct. Indeed, this approach would assume a very strange free energy profile F(x) with a discontinuous second derivative at the minimum. Rather, one should consider a Taylor expansion of F(x) of the form

F(x) = F(x0) + F′(x0)(x - x0) + (1/2) F″(x0) (x - x0)^2 + …

Since one can choose F(x) = 0 (a constant added to the energy does not change the physical description) and F’(x0) = 0 since x0 is an extreme (a minimum), we see that the quadratic term is the first non-zero term in the expansion (it represents the so-called harmonic approximation). Thus, the authors should fit just one parabola on both sides close to the minimum, and then see how the profile deviates from this parabola for larger deformations. The deviations may then very well be different for under- and overtwisting.

P. 4, 2nd line from above: isotropic ◊ homogeneous

P. 4: Equation 2 is incorrect. In fact, the twist persistence length and twist stiffness are two different quantities, the first one being the decorrelation length of twist fluctuations. They are closely related for an intrinsically straight, homogeneous rod where twisting is decoupled from bending (twistable worm-like chain, or TWLC), where twist persistence length is twice the twist stiffness (see e.g. the review of Dohnalova and Lankas, WIRES 2021). To correct their presentation, the authors can just put C/k_BT instead of C in eq. 1, say that C is expressed in nanometers, and not talk about twist persistence length at all (and take eq. 2 out). Beware: in your equations, T means two different things!

P. 4: "A multivariate Ising model is used …" A citation is missing here. But more importantly, please describe in detail what you did. What kind of deformation energies - they are known from the umbrella sampling, no? So, say in all the detail what you used the Ising model for, which tool, with what parameters, what the exact protocol was, etc. This method is certainly not standard and we need all the information.

P. 4, Bioinformatics: corresponding ◊ corresponding to

P. 4, idem: the to the binding

P. 4, Results: impact ◊ impacts; towards same ◊ towards the same; paly ◊ play

P. 5: Supercoiling density in this case is (tw - tw_ref)/tw_ref, thus just 0.15, not 0.15 per bp step.

P. 5, B-DNA integrity. It is not enough to monitor stretch and opening - for instance, many bp breaks proceed via shear. The simplest way (also for the reader) is to monitor the HB distances directly, e.g. as in Figure S8 of Dohnalova et al., Compensatory mechanisms in temperature dependence … JCTC 2020.

P. 5: The authors say that the restraint range a. is potentially exaggerated, b. represents an approximation to the changes, and c. these changes are not known. Instead, please provide a serious discussion about the extent of the torsional strain during DNA transcription, including literature references, and relate it to your choice.

P. 5: Average twist per bp step. There seem to be two different meanings of the word "twist". It may mean the end-to-end twist defined between the restraining handles, or a local twist of individual bp steps. It is not clear what the authors mean here, please clarify. In this and other places, it would be useful to employ different notions, e.g. end-to-end twist, as opposed to local twist (or twist for short) - see e.g. Dohnalova and Lankas, WIRES 2021.

P. 5: Increased torsional rigidity. The observed increase depends on the length of the DNA under investigation. For instance, in the limit of very long DNA fragment, rigidifying its short part will have almost no effect on the overall stiffness. Here, the quantitative comparison with MafB is not quite fair, since MafB covers all the restrained region (see Figure 4 of Hörberg and Reymer, Sci. Rep. 2020), while here there are 4 bp (nearly) protein-free at each side.

P. 5: Structural effects of twisting. The authors provide this information for (presumably local) twist changes very nicely in Fig. 2. Figures analogous to Fig. 2 but for the other conformational parameters (roll, slide, …) should be added to SI. Are the reference values in Fig. 2 different in the four parts, or are they always those of the naked DNA? It is very interesting that the changes in Fig. 2 are sometimes not monotonic, e.g. in MadMax the twist of the 3rd step (AG first increases, then decreases and then stays the same). Is it real, or just a lack of convergence (e.g. due to flickering protein contacts)? Similarly for the central CG of MaxMax.

P. 5 bottom: Deformation energy, Ising model. It is not clear at all what is going on here, please specify in detail. The deformation energies in Figure 1 have large error bars (with as yet unclear meaning) which mostly overlap each other, so they should be interpreted with caution. Please clarify.

P. 6, Figure 1: Parts B and C are interchanged.

P. 6, steps excluded from the analysis: these steps are present in Figures 2 and S8-S13. Perhaps they could stay there.

P. 7, "behaviour of other parameters follow earlier reports": which reports and in what sense?

P. 7, axis bending (not axes or ax): How is it defined? The bending in Fig. 3 refers to individual pairs, while Fig. S4 seems to show bending for the whole region. Please specify. Include units in Fig. 3.

P. 7, bending towards major/minor groove. A mere bending angle (Figs. 3 and S4) does not say anything about bending direction. How was the direction determined?

P. 7-8: The last sentence of the subsection is incomprehensible, please reformulate. Fig. 4: include units.

P. 9-10, protein-DNA interactions: It would be fair to mention explicitly that in some cases the protein interacts also outside the E-box, i.e. with the flanking sequence, which can then affect the torsional response of these regions.

P. 11, Discussion: see the beginning of this report. I would also suggest to put the bioinformatics findings into a new short subsection within Results, and leave the Discussion for their interpretation.

Supplement: Pay attention to the readability of the figures, inclusion of units, more complete descriptions in the captions of what is in the figure. I suggest to put Figures S8 to S13 each at one page in the landscape format. The current ones are nearly unreadable.

---

## [Reviewer Report]

*Comments to Author*: Reviewer #1: Hörberg et al. study structural effects of torsionally deforming DNA double helix with different transcription factors (TFs) bound to it by means of extensive umbrella sampling molecular dynamics (MD) simulations. In the cell, DNA is torsionally stressed during transcription, which comprises a major regulatory force in transcription regulation, operating via different mechanisms. In this work the authors demonstrate that in the presence of different TFs, the torsional stress induces different local structural changes to the DNA. The authors then hypothesize that these differences might have functional implications for transcription control, thus representing yet another regulatory mechanism, and give hints to support this view.

The work proceeds much along the lines of the authors’ prior publication (Hörberg and Reymer, Sci. Rep. 2020), where the hypothesis has already been formulated. In the present manuscript, the authors apply their approach to three more protein-DNA complexes with important TFs. Another innovation is that now the torsionally restrained region is wider than the protein binding site (4 flanking base pairs at each side), so that the effect on the flanking DNA can also be monitored.

My main concern is the following. The authors apply their torsional restraint and see TF-dependent structural changes within the protein binding site and the +/- 4 flanking pairs. Then they perform bioinformatics analysis using ChIP-seq data and find that each TF has different co-binders within +/- 100 bp around its binding site. The authors then hypothesize that the two phenomena are related, in particular they propose that torque-induced, protein-specific DNA deformations produce binding sites for the distinct co-binders (p. 11). However, the authors do not propose any plausible physical mechanism whereby the localized structural changes propagate as far as 100 bp away. There seems to be nothing in their data pointing to this possibility. The authors then focus on a specific example, the TBP and E2F factors. They mention (p. 11) that TBP and E2F deform DNA in a similar fashion as the one they observe, but no details are given, so it is hard to understand what they mean. Notice that e.g. TBP binds to the TATA-box (TATAAAA and its variants) which is 7 bp long, longer than the 4 flanking pairs modelled here. The authors should clarify these points.

Other issues:

Multiple times throughout the manuscript: use "respectively" instead of "correspondingly". Also, pay attention to the proper form of citations in the text ("Hörberg and Reymer" instead of "Johanna Hörberg and Reymer", etc.)

P. 2: The discussion about twist-absorbing dinucleotides is incomplete. Notice that if an inhomogeneous rod is torsionally stressed (below its buckling point), then its more flexible parts absorb more twist. This can be understood already in the framework of linear elasticity, by computing minimum energy twist distribution under the constraint of fixed total twist. The possible multiwell free energy landscape may (or may not) contribute to this, but multimodality is certainly not a prerequisite for the phenomenon. Please refer e.g. to a recent review on DNA elasticity (Dohnalova and Lankas, Wiley Interdiscip. Rev. Comput. Mol. Sci. 2021).

P. 2, bottom: The authors promise to investigate the relative positions of the TFs with respect to the nearest nucleosomes and transcription start sites, but nothing of this kind seems to be present in the manuscript.

P. 2, 5th line from below: as well ◊ as well as

P. 3, 2nd para: The restrained sets … ◊ The restraint sets …

P. 3, 2nd para: degrees^{-2} ◊ degree^{-2}

P. 3, MD simulation protocol: My understanding of the authors’ method to impose twist is that three base pairs at each side of the examined region are chosen to define the restraint. Please say which base pairs were chosen. Also say from which to which pair you define the twist between the handles (end-to-end twist). Perhaps it is defined between the middle pairs of each handle?

P. 3: It is very important to indicate the value of twist_ref and how it was obtained. Was twist_ref the same for all systems?

P. 3: There seem to be 10 windows for undertwisting and 10 for overtwisting, plus one for twist = twist_ref. That would make 10.5 microseconds, but you talk about 11.1 microseconds, please explain. Use the proper symbol for a microsecond.

P. 3-4, Elastic force constant analysis: The authors use a quadratic fit in the vicinity of the minimum for undertwisting and overtwisting individually. This does not seem to be correct. Indeed, this approach would assume a very strange free energy profile F(x) with a discontinuous second derivative at the minimum. Rather, one should consider a Taylor expansion of F(x) of the form

F(x) = F(x0) + F’(x0)(x - x0) + (1/2) F’’(x0) (x - x0)^2 + …

Since one can choose F(x) = 0 (a constant added to the energy does not change the physical description) and F’(x0) = 0 since x0 is an extreme (a minimum), we see that the quadratic term is the first non-zero term in the expansion (it represents the so-called harmonic approximation). Thus, the authors should fit just one parabola on both sides close to the minimum, and then see how the profile deviates from this parabola for larger deformations. The deviations may then very well be different for under- and overtwisting.

P. 4, 2nd line from above: isotropic ◊ homogeneous

P. 4: Equation 2 is incorrect. In fact, the twist persistence length and twist stiffness are two different quantities, the first one being the decorrelation length of twist fluctuations. They are closely related for an intrinsically straight, homogeneous rod where twisting is decoupled from bending (twistable worm-like chain, or TWLC), where twist persistence length is twice the twist stiffness (see e.g. the review of Dohnalova and Lankas, WIRES 2021). To correct their presentation, the authors can just put C/k_BT instead of C in eq. 1, say that C is expressed in nanometers, and not talk about twist persistence length at all (and take eq. 2 out). Beware: in your equations, T means two different things!

P. 4: "A multivariate Ising model is used …" A citation is missing here. But more importantly, please describe in detail what you did. What kind of deformation energies - they are known from the umbrella sampling, no? So, say in all the detail what you used the Ising model for, which tool, with what parameters, what the exact protocol was, etc. This method is certainly not standard and we need all the information.

P. 4, Bioinformatics: corresponding ◊ corresponding to

P. 4, idem: the to the binding

P. 4, Results: impact ◊ impacts; towards same ◊ towards the same; paly ◊ play

P. 5: Supercoiling density in this case is (tw - tw_ref)/tw_ref, thus just 0.15, not 0.15 per bp step.

P. 5, B-DNA integrity. It is not enough to monitor stretch and opening - for instance, many bp breaks proceed via shear. The simplest way (also for the reader) is to monitor the HB distances directly, e.g. as in Figure S8 of Dohnalova et al., Compensatory mechanisms in temperature dependence … JCTC 2020.

P. 5: The authors say that the restraint range a. is potentially exaggerated, b. represents an approximation to the changes, and c. these changes are not known. Instead, please provide a serious discussion about the extent of the torsional strain during DNA transcription, including literature references, and relate it to your choice.

P. 5: Average twist per bp step. There seem to be two different meanings of the word "twist". It may mean the end-to-end twist defined between the restraining handles, or a local twist of individual bp steps. It is not clear what the authors mean here, please clarify. In this and other places, it would be useful to employ different notions, e.g. end-to-end twist, as opposed to local twist (or twist for short) - see e.g. Dohnalova and Lankas, WIRES 2021.

P. 5: Increased torsional rigidity. The observed increase depends on the length of the DNA under investigation. For instance, in the limit of very long DNA fragment, rigidifying its short part will have almost no effect on the overall stiffness. Here, the quantitative comparison with MafB is not quite fair, since MafB covers all the restrained region (see Figure 4 of Hörberg and Reymer, Sci. Rep. 2020), while here there are 4 bp (nearly) protein-free at each side.

P. 5: Structural effects of twisting. The authors provide this information for (presumably local) twist changes very nicely in Fig. 2. Figures analogous to Fig. 2 but for the other conformational parameters (roll, slide, …) should be added to SI. Are the reference values in Fig. 2 different in the four parts, or are they always those of the naked DNA? It is very interesting that the changes in Fig. 2 are sometimes not monotonic, e.g. in MadMax the twist of the 3rd step (AG first increases, then decreases and then stays the same). Is it real, or just a lack of convergence (e.g. due to flickering protein contacts)? Similarly for the central CG of MaxMax.

P. 5 bottom: Deformation energy, Ising model. It is not clear at all what is going on here, please specify in detail. The deformation energies in Figure 1 have large error bars (with as yet unclear meaning) which mostly overlap each other, so they should be interpreted with caution. Please clarify.

P. 6, Figure 1: Parts B and C are interchanged.

P. 6, steps excluded from the analysis: these steps are present in Figures 2 and S8-S13. Perhaps they could stay there.

P. 7, "behaviour of other parameters follow earlier reports": which reports and in what sense?

P. 7, axis bending (not axes or ax): How is it defined? The bending in Fig. 3 refers to individual pairs, while Fig. S4 seems to show bending for the whole region. Please specify. Include units in Fig. 3.

P. 7, bending towards major/minor groove. A mere bending angle (Figs. 3 and S4) does not say anything about bending direction. How was the direction determined?

P. 7-8: The last sentence of the subsection is incomprehensible, please reformulate. Fig. 4: include units.

P. 9-10, protein-DNA interactions: It would be fair to mention explicitly that in some cases the protein interacts also outside the E-box, i.e. with the flanking sequence, which can then affect the torsional response of these regions.

P. 11, Discussion: see the beginning of this report. I would also suggest to put the bioinformatics findings into a new short subsection within Results, and leave the Discussion for their interpretation.

Supplement: Pay attention to the readability of the figures, inclusion of units, more complete descriptions in the captions of what is in the figure. I suggest to put Figures S8 to S13 each at one page in the landscape format. The current ones are nearly unreadable.

Reviewer #2: This paper describes a fascinating new mechanism for supercoiled-controlled DNA recognition, whereby bound proteins trap supercoils, thereby changing the twist of the DNA upstream, and changing the affinity of this site for a second DNA binding protein. Simulations have been performed at various levels of DNA twist using restrained MD in the presence of three related DNA binding proteins, and the sequence dependent absorption of the twist imposed carefully assessed, thereby demonstrating how these different proteins impede supercoiling. The simulations performed have been carefully described and the analysis thoughtfully executed, with detailed information including error bars presented as Supplementary Information. The addition of Chip-Seq analysis suggests the potential for binding partners of the proteins investigated to co-operatively associate in a manner that can be controlled by supercoiling. I find this proposed mechanism very compelling, and therefore recommend publication of this interesting paper.

I have one recommendation to improve the readability of the paper. The authors may consider including the full proteins investigated in Figure 1, ideally highlighting differences in sequence, as currently the reader only gets to see this information when they reach the section on DNA protein contacts, or if they look at Supplementary Figure 2 (which explains this very clearly). They may also wish to carefully check the legend to Figure 1, as I found the numbering and lettering somewhat confusing.